# Mechano-Chemistry across Phase Transitions in Heated Albumin Protein Solutions

**DOI:** 10.3390/polym15092039

**Published:** 2023-04-25

**Authors:** Chingis Kharmyssov, Kairolla Sekerbayev, Zhangatay Nurekeyev, Abduzhappar Gaipov, Zhandos N. Utegulov

**Affiliations:** 1Department of Physics, School of Sciences and Humanities, Nazarbayev University, 010000 Astana, Kazakhstan; 2Science Department, Astana IT University, 010000 Astana, Kazakhstan; 3Institute for Experimental Physics, Hamburg University, Luruper Chaussee 149, 22761 Hamburg, Germany; 4Department of Medicine, School of Medicine, Nazarbayev University, 010000 Astana, Kazakhstan

**Keywords:** Brillouin scattering, Raman scattering, viscoelastic, albumin, phase transitions, heating, denaturation, gelation, mechano-chemical, protein, polymer

## Abstract

The presence of certain proteins in biofluids such as synovial fluid, blood plasma, and saliva gives these fluids non-Newtonian viscoelastic properties. The amount of these protein macromolecules in biofluids is an important biomarker for the diagnosis of various health conditions, including Alzheimer’s disease, cardiovascular disorders, and joint quality. However, existing technologies for measuring the behavior of macromolecules in biofluids have limitations, such as long turnaround times, complex protocols, and insufficient sensitivity. To address these issues, we propose non-contact, optical Brillouin and Raman spectroscopy to assess the viscoelasticity and chemistry of non-Newtonian solutions, respectively, at different temperatures in several minutes. In this work, bovine and human serum albumin solution-based biopolymers were studied to obtain both their collective dynamics and molecular chemical evolution across heat-driven phase transitions at various protein concentrations. The observed phase transitions at elevated temperatures could be fully delayed in heated biopolymers by appropriately raising the level of protein concentration. The non-contact optical monitoring of viscoelastic and chemical property evolution could represent novel potential mechano-chemical biomarkers for disease diagnosis and subsequent treatment applications, including hyperthermia.

## 1. Introduction

The concentration of biomacromolecules in biofluids is an important biomarker for various disease states. One example is the concentration of hyaluronic acid in synovial fluid, which can indicate the joint grade and provide insights into conditions such as rheumatoid arthritis [1] and periprosthetic joint infection [2]. Changes in the level of hyaluronic acid have been linked to the inhibition of antithrombin and consequent joint tissue deterioration in rheumatoid arthritis [3]. A decrease in the concentration of hyaluronic acid is also associated with a worsening of rheumatoid arthritis [4]. Therefore, monitoring the concentration of hyaluronic acid is crucial for making informed clinical decisions. Other biofluids also have biomarkers linked to disease states, such as elevated fibrinogen protein in blood plasma being related to the occurrence and progression of cardiovascular disorders [5,6,7]. The concentration of tau protein and amyloid beta in cerebrospinal fluid has been associated with the onset of Alzheimer’s disease [8]. In addition, protein concentration is a major marker of chronic kidney diseases [9,10,11,12].

Examining how phase transitions affect the dynamics and structure of biopolymers has been identified as an important biomarker for various disease states [13,14]. It is well-known that highly organized assemblies of globular proteins are associated with neurodegenerative disorders. Studies on proteins [15], DNA [16], RNA [17], and polysaccharides [18] have demonstrated that structural and dynamic alterations to biomolecules can have a significant impact on how well they operate. A protein must unfold, at least partially, from its native state for aggregation to take place, because each polypeptide chain is more likely to form intermolecular connections when it is unfolded or partially folded, which reduces contact with the solvent [19]. This enables the creation of substantial protein aggregates whose structure is greatly influenced by the medium’s overall properties (polarity, acidity, ionic strength, etc.), which can strengthen intermolecular interactions [20,21]. An irreversible denaturation process is seen when clusters are formed, and a cooperative rearrangement can eventually produce organized oligomers (e.g., beta sheets) [22]. The reversible denaturation of a single molecule is tied to the overall unfolding of the polypeptide chain if intermolecular interactions are stimulated and secondary reactions are prevented. It is important to note that the rates of reversible and irreversible denaturation differ significantly, and that extended incubation times are typically necessary to detect the development of large clusters and, eventually, gel formation [23]. Usually, unfolding moves considerably more quickly [24].

Various techniques are used to measure the concentration of macromolecules in biofluids, but they often require large sample sizes or take several hours to produce results. Traditional protein assays require several steps and specialized equipment, meaning that they are time-consuming and necessitate expertise in protein handling [25].

An alternative method is to indirectly measure protein concentrations through rheological properties such as shear viscosity and the longest relaxation time. Biofluids, which are mainly composed of water, become non-Newtonian when macromolecules are added, causing the solution to have additional viscous and elastic properties. According to the well-established relations in polymer physics [26], the shear viscosity measures the drag exerted by the biofluid under external flow conditions, while the longest relaxation time measures the elasticity stored within the suspended macromolecules. This concept has been used to identify potential biomarkers for several diseases, including cardiovascular diseases [5,6] and joint infections [2]. Brillouin spectroscopy provides a non-invasive and label-free alternative for investigating the viscoelastic properties of living matter. Originally developed in material science, Brillouin spectroscopy has been successfully applied in the field of material science [26,27], enabling the direct imaging of viscoelastic properties in 3D and at high-resolution using scanning confocal microscopy. Brillouin scattering is a sensitive technique that can detect small changes in the storage modulus and large changes in the loss modulus of diluted samples due to hydrophobic hydration. It can also measure microviscosity and detect the early stages of the transition from the liquid to the solid state in polymer samples [28]. We employed Brillouin and Raman spectroscopy for the temperature-dependent mechano-chemical measurement of non-Newtonian solutions. Recent advancements in Brillouin light scattering (BLS) spectroscopy [29] have yielded useful information on viscoelastic properties [30,31], phase transition, and collective dynamics within interrogated condensed matter, such as biopolymers [32], cells [33], and tissues [34] due to the probing of inelastically scattered laser light from GHz acoustic phonons in the studied medium [35,36]. Techniques such as fluorescence, circular dichroism, and standalone Raman spectroscopy can address the structural characterization of proteins on a chemical level, but they are unable to shed light on the macroscopic viscoelastic properties of gel-like systems provided by BLS spectroscopy, which enables collective analysis to address denaturation and gelation processes in globular protein solutions [37]. The shift in the Brillouin peak is proportional to the storage (elastic) modulus, while its spectral peak width is proportional to the loss (viscosity) modulus of the optically interrogated medium.

Research on protein phase transitions in the high-temperature zone has mostly been conducted on a single-domain lysozyme with a suitably straightforward structure using inelastic light scattering [38]. However, lysozyme is a model protein with a simple structure, and exploring further the dynamics of the phase transition of proteins with more complex proteins would be helpful. Human serum albumin (HSA) and bovine serum albumin (BSA) are more complex model proteins that are present mainly in the blood plasma of mammals [39]. Albumin is critical for sustaining the oncotic pressure necessary for the suitable exchange of body fluids between blood vessels and body tissues. While both albumins have similar primary structures, the amino sequence alignment of HSA presents about 76% sequence identity with BSA [40]. Both proteins are identical in terms of some physical criteria, for example, their partitioning behavior in an aqueous two-phase system and surface hydrophobicity [41]. However, there are slight differences concerning their thermal and chemical stability [42], electrophoretic behavior [43], and binding and photochemical properties [42,44]. However, due to its accessibility, BSA is frequently utilized in scientific investigations instead of HSA, with the results being translated over to HSA [45]. While there are many characteristics that the two homologs have in common, there are also significant distinctions, such as the notion that HSA is more hydrophobic and thermally stable than BSA [46,47].

Comparative studies of HSA and BSA solutions have been carried out employing different experimental techniques [48,49]. However, there are significant open questions regarding the viability of isolating the different effects due to the delicate competition between reversible and irreversible denaturation. Additionally, because of the varying sensitivity of the experimental procedures, it is frequently challenging to characterize structural characteristics completely, from the changes in various side chains to the macroscopic arrangement of supramolecular species.

In this work, temperature-dependent Brillouin and Raman spectroscopic measurements were applied to HSA and BSA aqueous solutions to characterize the thermally induced unfolding and aggregation at different protein concentrations. We aimed to provide in-depth knowledge regarding the alterations in the physicochemical characteristics of albumins driven by heat over a wide range of protein concentrations.

## 2. Materials and Methods

Protein powder samples were stored in a refrigerator at −18 °C. HSA and BSA were obtained from Sigma-Aldrich as a lyophilized powder with 96% purity and used without any further purification. A working solution of the proteins was dissolved in phosphate-buffered saline (PBS). One tablet of phosphate buffer was dissolved in 200 mL of deionized water yielding 0.01 M phosphate buffer, 0.0027 M potassium chloride, and 0.137 M sodium chloride at pH 7.4 and 25 °C. The weight concentrations were 1%, 5%, 10%, 15%, and 20% for both proteins. The solution was then placed into a temperature-controlled cavity well of a glass slide, covered with cover glass, and sealed with silicon oil to prevent water evaporation. The sample temperature was controlled by a thermoelectric heating stage (Linkam PE120) with a set temperature value; then, the sample temperature was stabilized for 3 min, and the BLS and Raman spectra were collected for the next 3 min. All the above-mentioned steps were repeated three times to obtain the standard error.

BLS spectra were measured with a single longitudinal-mode continuum wave 532 nm Coherent Verdi-G2 laser with an incident laser power of 10 mW focused on protein solution samples contained in glass cavity slides with a 5× microscope objective. The Brillouin and Raman backscattered light from protein solutions were separately spectrally analyzed by Sandercock scanning using a six-pass Tandem Fabry-Perot interferometer (Table Stable TFP-2) and by a confocal LabRAM HR Evolution (Horiba) Raman microscope, respectively.

## 3. Results and Discussion

### 3.1. Brillouin Scattering of BSA and HSA Solutions

The Brillouin spectra of the studied protein solutions contained a single longitudinal acoustic peak measured over the 60–98 °C temperature range with a 2 °C increment. The Brillouin spectral peaks with fitted Voigt profiles are presented in Figure 1. In this work, only BLS data at T > 60 °C are presented, since the spectral behavior within the 20–60 °C temperature interval was featureless and had the same trend for both protein solutions as well as for the host PBS solution and distilled water.

Figure 2a shows a comparison of the temperature-dependent Brillouin peak shift positions associated with elasticities (storage modulus) at different concentrations of HSA and BSA solutions, as well as pure water. The data obtained for the solutions were of course different from those of pure water and PBS due to their different compressibility, densities, and viscosities. However, at these temperatures and wave vectors, the solvent’s characteristics largely determine how binary systems react to acoustic excitation. At T = 60 °C –70 °C, with the rise in protein concentration, both protein solutions became generally stiffer, in line with previous observations [50,51,52]. All experiments were carried out with PBS; therefore, the addition of salt reduced the electrostatic repulsion between proteins and affected the protein gel microstructure and aggregation [52]. Heat exposure can cause proteins to aggregate and form a gel if the protein concentration is high enough. The denaturation process exposes hydrophobic residues that interact with neighboring molecules to form aggregates and gel-like structures. The protein concentration is a critical factor in determining the type of structure formed during heating. The resulting aggregation under denaturation leads to gel structures forming. The decreasing coarseness results in thinner strands and thereby a decrease in stiffness. However, when amino acid strands decrease in thickness, a progressively increasing number of connectivity nodes are made accessible due to the large amount of protein. These changes in the protein microstructure increase the stiffness. The BSA solutions were observed to be stiffer than the HSA solutions for all temperatures and became even stiffer with a rise in protein concentration.

The difference in stiffness between the HSA and BSA solutions could also be attributed to their conformational and colloidal stabilities. Dynamic light scattering data have shown that HSA has a greater solution-phase conformational stability than BSA. This means that HSA molecules tend to maintain their folded structure and are less likely to denature or aggregate in solution, resulting in a more rigid and ordered solution structure. However, despite its higher conformational stability, HSA has been found to have lower colloidal stability than BSA [53]. This means that HSA molecules are more prone to aggregating or sticking together in solution, which can lead to a loss in stability and an increase in solution viscosity. On the other hand, BSA has higher colloidal stability, meaning that its molecules are more resistant to aggregation and precipitation, resulting in a more stable and uniform solution structure [54]. Overall, the difference in stiffness between the HSA and BSA solutions could be attributed to their structural differences and resulting differences in binding forces, as well as their conformational and colloidal stabilities. These factors could affect how the protein molecules interacted with each other and with the solvent molecules in the solution, influencing the overall stiffness and stability of the solution.

One of the structural changes that can take place in proteins is denaturation, which is the loss of their native structure. For the 1% BSA and HSA solutions, the temperature-dependent Brillouin peak spectral position behavior was featureless and almost identical. Earlier differential scanning calorimetry studies showed that at 65 °C, BSA starts to unfold in 0.01 M phosphate [55]. According to our BLS data, at 5%, both albumin protein solutions transitioned into an irreversible denaturized state at $T_{den}^{5\%}\left(BSA\right)=T_{den}^{5\%}\left(HSA\right)=67\;{}^{\circ}C$, whereas, at 10%, the denaturation temperatures rose to $T_{den}^{10\%}\left(BSA\right)=68\;{}^{\circ}C$ and $T_{den}^{10\%}\left(HSA\right)=72\;{}^{\circ}C$ for the BSA and HSA solutions, respectively. After reaching their respective denaturation temperatures, both protein solutions typically lost stiffness, as evidenced by the decrease in the corresponding Brillouin shifts with a further temperature rise. This loss of post-denaturation elasticity became especially evident with the rise in protein concentration, especially for the HSA solution. The observed denaturation temperature for the BSA solution was in agreement with earlier BLS measurements, which exhibited protein denaturation at $T_{den}^{10\%}$ > 60 °C [56,57].

At $T>T_{den}$, albumin proteins tend to form gels that are defined by a specific protein molecular structure; interaction with other components (salts, acids, urea, etc.); and processing conditions such as the heating and cooling rate, temperature, pH, and ionic strength [58]. As for the gelation temperature, it starts to increase from $T_{gel}^{5\%}\left(BSA\right)=82\;{}^{\circ}C$ to $T_{gel}^{5\%}\left(BSA\right)=90\;{}^{\circ}C$ for BSA solutions and from $T_{gel}^{5\%}\left(HSA\right)=77\;{}^{\circ}C$ to $T_{gel}^{10\%}\left(HSA\right)=94\;{}^{\circ}C$ for HSA solutions. Therefore, during the heating of the protein solutions, the increased protein concentration caused increases in the denaturation and gelation temperatures, i.e., a higher concentration of proteins in the PBS solution delayed both structural phase transitions associated with denaturation and gelation upon gradual heating. At a 5% weight concentration, the HSA solution appeared to be more resistant to thermally induced denaturation than the BSA solution, while the trend flipped at a 10% weight concentration.

Let us examine the Brillouin light scattering results for the 10% protein solution within the context of the temperature-dependent phase transitions listed in Figure 3, which were based on existing literature. There were no unusual patterns in the temperature-dependent behavior of the Brillouin doublet shift, FWHM, and intensity observed in the temperature range below 57 °C that corresponded to reversible conformational transitions. The HSA solution again appeared to be more resistive than the BSA solution, but this time to thermally induced gelation. BSA and HSA underwent a similar series of phase transitions that could not be reversed when heated above a temperature of 57 °C and 62 °C, respectively, causing aggregations. The denaturation region of BSA is typically identified by anomalous behavior in the heat capacity observed within the range of 57–81 °C [59]. Inelastic neutron spectroscopic experiments also revealed anomalies in the behavior of the mean square displacement in this temperature range [60]. The thermal denaturation of BSA leads to unfolding, structural changes, and aggregation. Studies using Raman spectroscopy, Fourier-transform infrared spectroscopy (FTIR), and other techniques have shown that denaturation results in the rearrangement of BSA’s secondary structures [61,62,63], causing the protein to lose its native properties. FTIR spectroscopy spectra revealed that BSA aggregation during denaturation was accompanied by changes in both the secondary structure (the loss of α structures and the formation of β layers) and the tertiary structure [62]. The inelastic neutron spectroscopy analysis of the relaxation processes observed during denaturation revealed two relaxation mechanisms depending on the BSA concentration in the solution [59]. Small-angle X-ray (SAXS) and small-angle neutron (SANS) scattering studies revealed the formation of aggregates at the beginning of BSA denaturation and the growth of their dimensions, radii of gyration, and molecular weights [64,65,66]. Typically, a melting point within the denaturation region is identified, with the temperature determined by the maximum in the heat capacity anomaly. This temperature is dependent on experimental conditions such as the pH and concentration. For high concentrations, sodium phosphate buffer, and pH > 7, the self-assembly aggregation temperature (melting temperature, T_m_*)* is around 62 °C [67]. Heating the BSA protein to its T_m_ resulted in the irreversible unfolding of domain II, followed by the unfolding of domain I with further heating. In the temperature region around 70 °C, SAXS studies showed the formation of burst-like high-order aggregates that transformed into rod-like structures, protofibrils, and eventually β-sheet-rich mature fibrils during morphological transitions, as observed via an electron microscope [68].

This research involved the use of SAXS and SANS to examine the behavior of BSA during denaturation [64,65,69]. The study showed the formation of aggregates at the start of BSA denaturing and their growth in size, weight, and dimensions. The maximum heat capacity anomaly was used to determine the melting point of denaturation, which could vary depending on the experimental conditions. The changes in the BSA structure during denaturation were found to be irreversible and resulted in the unfolding of different domains. As the temperature increased, protein aggregation was observed in the form of high-order aggregates, which transformed into various structures such as rod-like structures, protofibrils, and fibrils.

Another temperature region in the denaturing region was identified, beginning at around 70–81 °C, where a gel-like phase was formed due to cross-linking between the proteins. This cross-state region was characterized by a fractal dimension that increased with a further increase in temperature after gel transitions up to saturation. The alterations in the spectra of the HSA solutions, which depended on the concentration, could have been due to various factors. These included changes in the population density of HSA resulting from a change in concentration, alterations in the secondary structure of HSA, and changes in the surrounding environment of HSA’s side chains. Another possible factor was changes in hydration. These deviations suggested the likelihood of alterations in the secondary structure of the protein, modifications in the microenvironment of HSA’s side chains, and adjustments in hydration. It was evident that the concentration-dependent changes in a protein significantly affected its hydration. As the concentration increased, there was an increase in the number of bonded water molecules and a decrease in the number of free water molecules. When the protein concentration in the solution was increased at temperatures between 70 °C and 81 °C, the measured Brillouin shift and width values were substantially enhanced. This indicated a considerable change in the protein solution’s structural properties, as observed by BLS. When the protein concentration was sufficiently high, the protein solution further aggregated, forming a gel at ~81 °C. We noted that gelation could only be observed with a protein concentration of at least 10%.

The temperature dependence of the Brillouin peak widths associated with the viscous (loss modulus) behavior of the HSA and BSA solutions is shown in Figure 2b. The viscous behaviors were similar for lower protein concentrations. An increase in viscosity was observed with an increase in protein concentration, which was in agreement with theories that account for an increase in inter-protein interaction potential and excluded the volume of water that was applied in predicting the increase in viscosity with the protein concentration [70,71,72,73,74]. We could observe that the 1% BSA and HSA solutions in PBS were almost as viscous as the PBS solution itself. However, at larger protein concentrations, both protein solutions became more viscous than the pure PBS solution, especially at higher temperatures. Interestingly, the 10% and 20% HSA solutions became more viscous than the BSA solutions at the same concentrations and showed the opposite effect to the elasticity trends of both protein solutions observed in Figure 2a.

### 3.2. The Secondary Structure

The relative variations in the intensities of the Raman spectra of albumin measured at different temperatures provide information about the structural degradation of the proteins. Temperature-dependent Raman spectroscopy on BSA solutions has shown that the onset of protein reversible conformational changes takes place at 42 °C. The partially reversible unfolding temperature region is between 50 °C and 60 °C. Irreversible denaturation and gelation take place at 60 °C and 70 °C, respectively [75]. This was also confirmed by cooling experiments conducted by Sassi et al. [37], who reported that the temperature profiles obtained from the three experiments were almost identical within the temperature range of 20–70 °C.

The Raman spectra of 5, 10, 15, and 20% HSA and BSA solutions in PBS were measured under heating in the 20–98 °C range with a 3 °C temperature increment, 80 s acquisition time, 3 accumulations, and 532 nm wavelength laser excitation. HSA and BSA are exemplary α-proteins as they contain a large number of α-helix structures and zero β-sheets [76,77,78,79]. We focused on the Raman peaks at 938 cm^−1^ and 1246 cm^−1^, attributed to an α-helix ordered structure [75,80] and amide III region random structure [81], respectively. The Raman spectrum peaks’ temperature evolution is shown in Figure 4 (938 cm^−1^) and Figure 5 (1200–1360 cm^−1^). The intensity of the 938 cm^−1^ peak was indicative of the ordering degree of the protein structures within the PBS solution. The 938 cm^−1^ peak was associated with skeletal stretching mode ν(C-C-N)_sym_ of an α-helix structure [75,80,82]. The decrease in its intensity with the temperature rise indicated an α-helix structure concentration decrease. Noticeable changes in the 938 cm^−1^ band occurred after 59 °C and 74 °C in the BSA and HSA, respectively.

The Raman peak intensity of the 1246 cm^−1^ vibrational band was associated with random structure increases along with the temperature rise. A more accurate analysis of the temperature behavior of the 1246 cm^−1^ band could be performed by considering the intensity of the reference peaks. Figure 6 shows the temperature dependence of the peak intensity ratios of 1246 cm^−1^ to 1337 cm^−1^ (I_1246/1337_) for the BSA and HSA solutions in PBS at various protein concentrations ranging from 5 to 20%. The I_1246/1337_ ratio characterized the degree of the protein structure randomization. The Raman peak signal of the 1% solution was too weak to be detectable. In the BSA and HSA solutions, the I_1246/1337_ ratio generally increased with the temperature rise, which was attributable to overall protein denaturation and randomization, respectively, with progressive heating at T > 60 °C.

In the HSA solution, the I_1246/1337_ ratio was lower compared to that of the BSA solution. It could be concluded that the HSA solution was more resistive to thermal denaturation than the BSA solution for all measured solution concentrations. This trend was in agreement with the measured Brillouin viscous (peak width) behavior. Therefore, chemical transitions within the alpha-helix and amid III region were likely correlated with viscosity trends, as opposed to stiffness trends.

### 3.3. Side-Chain Exposure and Arrangement of Tertiary Structure

A spectral region at 450–700 cm^−1^ was also observed in the case of a protein concentration of 20%. In BSA and HSA, which contain multiple disulfide bonds (S–S stretching mode), this mode could be observed in the Raman spectra as a characteristic band at around 510 cm^−1^. When proteins are subjected to thermal treatment, changes in their tertiary structure occur, which can result in the exposure of side chains to the solvent and modifications to sulfur moieties [83]. These changes are indicated by the appearance of aggregation and gelation. Under non-reducing conditions, such as the absence of denaturing agents such as urea or tetramethyl urea, disulfide bonds remain intact, and the hydrophobic effect induced by short-range protein–protein connections, such as van der Waals forces, H-bonding, and opposite charge interactions, reduces the exposure to the solvent. To track these effects during denaturation, the Raman spectra of the solutions were analyzed for the marker band of disulfide bonds, which corresponded to the S-S stretching mode. BSA and HSA solutions are considered intriguing models for investigating protein unfolding and refolding processes because they possess up to 17 disulfide bridges each [84]. The Raman measurements were performed for HSA and BSA under the same conditions as the Brillouin experiments. The results are shown in Figure 7. The decline in the intensity of the 510 cm^−1^ band at elevated temperatures, as shown in Figure 7, indicated that the S-S bonds were breaking due to unfolding. HSA-susceptible S-S bonds are more likely to break at temperatures above 80 °C. One possible reason why S-S bonds in HSA are more susceptible to breakage is that they are more exposed and accessible to the surrounding environment. S-S bonds are usually located on the surface of the protein, where they can be exposed to solvent molecules, ions, or other molecules that can affect their stability. In addition, susceptible S-S bonds may be located in more flexible regions of the protein, which can increase their mobility and susceptibility to breakage. The sudden increase in the intensity of the 510 cm^−1^ peak at 74 °C at this temperature may indicate the formation of new disulfide bonds.

## 4. Conclusions

Increasing the density of HSA and BSA protein solutions caused them to become stiffer and more viscous at low concentrations within the regimes of denaturation temperatures. BSA generally formed stiffer and more viscous solutions compared to HSA at low concentrations, while at higher concentrations this trend reversed. Upon reaching its denaturation temperature $T_{den}^{10\%}\left(HSA\right)=72\;{}^{\circ}C$, at the protein concentration of 10%, the HSA solution suddenly became elastically weak and simultaneously highly viscous. This trend was even more obvious with the largest protein concentration of 20%, which may be attributed to volume exclusion by the protein solution. The difference between BSA and HSA across phase transitions involving denaturation and gelation was attributed to the difference in thermal stability, chemical properties, and hydrophobicity. Proteins at temperatures above the denaturation point expose their inner hydrophobic parts and tend to group up to escape the hydrating water. This may lead to the formation of groups of proteins and further gelation, leading to sudden stiffening. Solutions with larger protein concentrations cause delays in denaturation and gelation upon heating.

Increasing the protein concentration in the HSA and BSA solutions typically led to the clear identification of the characteristic phase transitions associated with the onsets of the denaturation and gelation processes, as evidenced in sudden changes in the elastic and viscous properties, as well as in the chemical structure of the protein order.

The obtained spectroscopic results could have great implications for the practical control of structural phase transitions in biofluids in response to external stimuli such as heating via tuning their protein concentrations. Our results also open avenues for the further exploration of the high-frequency viscoelastic and chemical behavior of variable protein concentrations undergoing structural phase transitions at elevated temperatures. The presented approach could also be of practical significance for the monitoring of the viscoelastic properties of protein-containing flesh tissues under hyperthermia in theranostic applications [85].

## Figures and Tables

**Figure 1 polymers-15-02039-f001:**
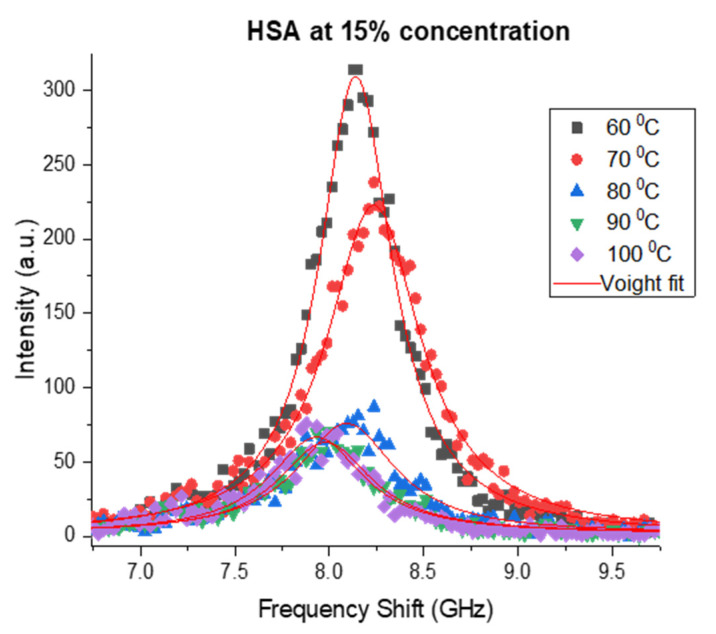
Temperature-dependent Brillouin spectra of 15% HSA solutions.

**Figure 2 polymers-15-02039-f002:**
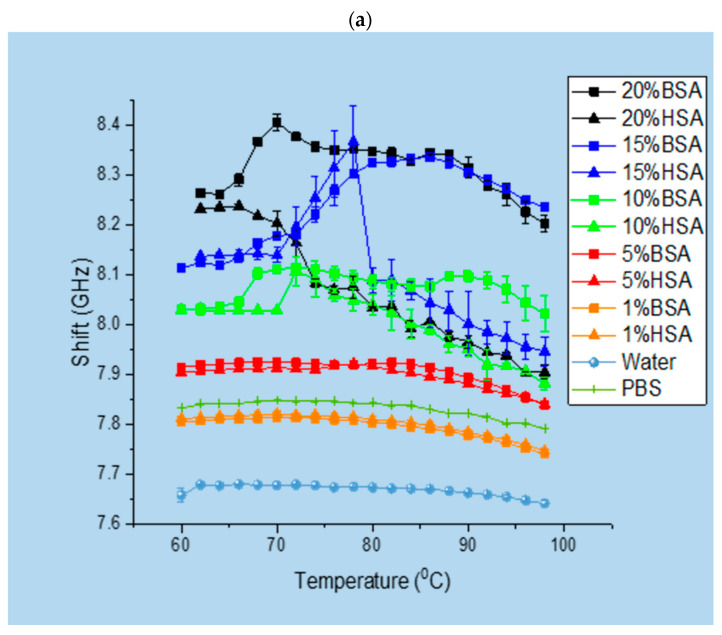
Brillouin spectroscopy. Dependencies of (**a**) the temperature of the Brillouin peak shift and (**b**) the Brillouin peak width (as the full width at half maximum—FWHM).

**Figure 3 polymers-15-02039-f003:**
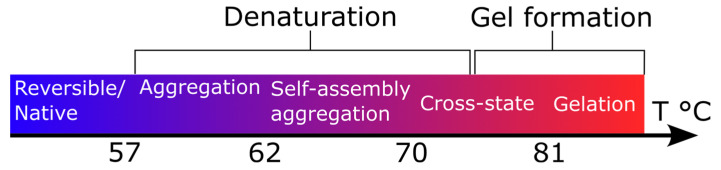
The sequence of phase transitions in serum albumin solution (pH = 7.45, protein concentration = 10%) [59,60,61,62,63,64,65,66].

**Figure 4 polymers-15-02039-f004:**
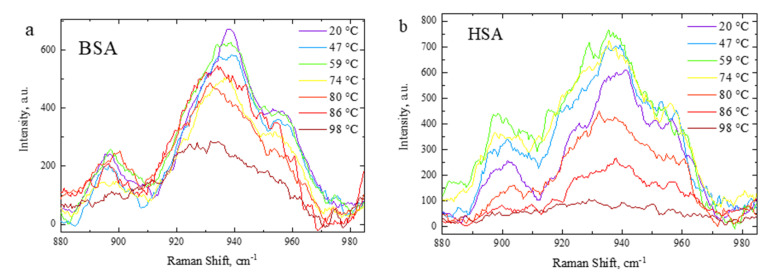
Temperature evolution of Raman spectra in the range 900–980 cm^−1^: (**a**) BSA and (**b**) HSA.

**Figure 5 polymers-15-02039-f005:**
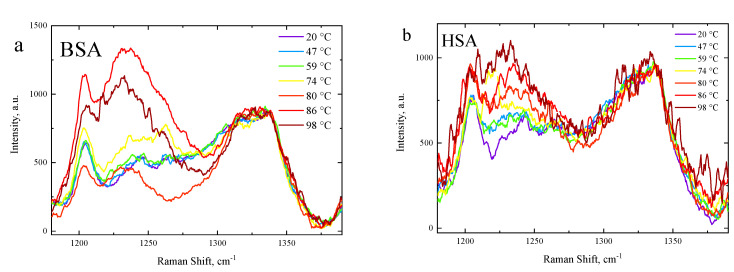
Temperature evolution of Raman spectra in the range 1200–1360 cm^−1^: (**a**) BSA and (**b**) HSA.

**Figure 6 polymers-15-02039-f006:**
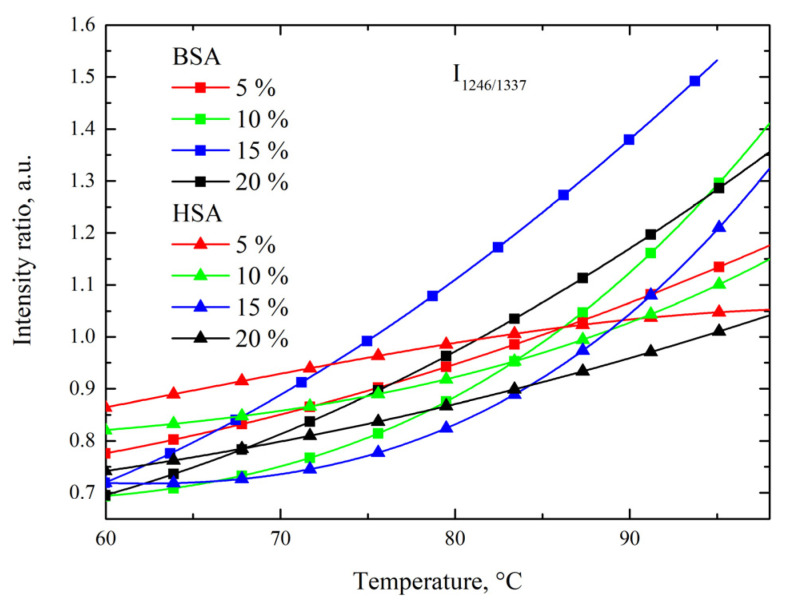
Temperature dependence relative peak intensity ratio of 1246 cm^−1^ band compared to 1337 cm^−1^ band of BSA and HSA protein solutions.

**Figure 7 polymers-15-02039-f007:**
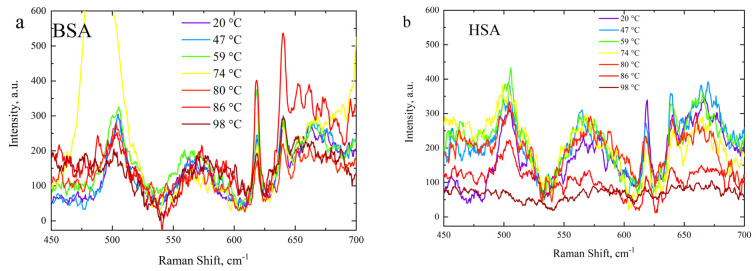
Temperature evolution of Raman spectra in the range 450–700 cm^−1^: (**a**) BSA and (**b**) HSA.

## Data Availability

The data are available by requesting the authors.

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
