# Peer review of "Mechano-Chemistry across Phase Transitions in Heated Albumin Protein Solutions"

_polymers, 2023, doi:10.3390/polym15092039_

Round 1
Reviewer 1 Report
This is a good work that evaluates the temperature-dependent Brillouin and Raman spectroscopic measurements that have been applied to the HSA and BSA aqueous solutions to characterize the thermally induced unfolding and aggregation at different protein concentrations. My specific observations are the following:
- Figure 3 and figure 4 are presented in wrong order and sense. Caption of figure 3 is not in agreement with figure, figure 3a indicates that is BSA and the caption indicates that is HAS. The same for figure 3b
- In the manuscript authors indicate that Figures 3 is for 938 cm-1 and figure 4 is for 1200 – 1360 cm-1, however figures are in the opposite scales.
- Instead of X°C, it should say X °C.
- Instead of cm-1, it should say cm-1
Reviewer 2 Report
The manuscript focus on a simple strategy based on Brillouin and Raman spectroscopy to monitor temperature-dependent mechano-chemical measurements of 2 HSA and BSA. It is an interesting and meaningful topic. However, the manuscript shows an imperfect study. The discussion about the results is too simple in the manuscript. The actual sample was deficient.
Reviewer 3 Report
In this study, the authors examine the behaviour of proteins HSA and BSA as a function of temperature using a combination of Brillouin and Raman spectroscopy. Brillouin spectrocscopy provides access to the rheological properties of the samples as they are heated , while Raman spectroscopy gives insight into changes in protein conformation. The novelty and utility of the study is being able to probe the viscoelastic properties as a function of temperature, as these are not easily accessible by other methods. Being able to probe temperature dependent rheological properties has applications outside biology, including foods and biopharmaceuticals. As such, I think the results of the paper are interesting, but I have major comments that need to be addressed below.
I found it very difficult to see how the authors draw conclusions about the gelation and denaturation temperatures based on the plots shown in Figure 2.
On, Lines 171 to 173 - The authors indicate they have deduced denaturation temperatures from their BLS data, but it is not clear to me why BLS is related to denaturation. Denaturation corresponds to protein unfolding, while BLS is being used to probe rheological properties? From examining Figure 2 for the data at 5% protein concentration, I do not see any transition at 67 C, which is mentioned as the unfolding temperature for BSA. Also the authors mention that the denaturation is irreversible, how do they know it is irreversible? Have they done any cooling experiments?
On lines 186 to 188 - The authors need to be clear about the criteria for determining the gelation temperature from the plots shown in Figure 2. Generally thie might be determined from the crossover between the storage and loss modulus, but these paramaters are not being measured directly?
Lines 235 to 238 - The authors are intrepreting the ratio I938/1000 as a direct measure of protein helix structure. However, this is very mis-leading. The results indicate this ratio changes with increasing protein concentration at room temperature implying changes to HSA or BSA conformation. This is physically unrealistic as these proteins do not change conformation at room temperature irrespective of protein concenteation. In addition , this ratio changes when increasin temperature from 20 to 40 C, which also cannot be due to changes in protein conformation. The authors need to consider other reasons for what these ratios depend upon if they are to be related to protein conformational changes at higher temperature.
Lines 244 to 246 The authors can not deduce that HSA is more stable against denaturation based upon considering the difference in the I938/1000 ratio for the reason state above. Usually the stability against aggregation is reflected by the unfolding temperature, maybe the authors should consider a comparison of the peaks in the I938/1000 profiles as a function of temperature, are these peaks related to unfolding temperatures?
Minor comments:
In Methods section, need to indicate that they have also studied proteins at concentrations of 15%.
Lines 170 to 171 - the authors reference a dynamic light scattering(DLS) study indicating BSA unfolds at 65 C. However DLS detects size changes and cannot discriminate between unfolding and aggregation so it is not possible to determine an unfolding temperature. The authors need to cite papers using techniques which directly probe unfolding such as differential scanning calorimetry, CD, or possibly fluorescence. In addition, unfolding temperature are sensitive to solution conditions, so for this study, the authors should cite studies measuing unfolding in PBS buffer.
Lines 157 to 158 is misleading. The authors are indicating that denaturation leads to coarser gel structures, but in the absence of denaturation, gel structures are not generally formed since proteins do not aggregate in the native state.
Paragraph between lines 233 to 239 should be combined with the previous paragraph since this paragraph is explaining why 1000 cm-1 is used as a reference which is diretly relevant for why the peak intensity ratio I938/I1000 is used as a measure of protein conformation.
Figure 3b - The plots do not appear to be normalized so it is very difficult to visualize how the peak intensity at 1246 cm-1 is changing with increasing temperature.
Lines 223 to 224 - Figure labels are mixed up. Figure 3 contains data for 1200 to 1360 cm-1, while Figure 4 contains data for 938 cm-1.
